# Interaction Between Human Skeletal and Mesenchymal Stem Cells Under Physioxia Enhances Cartilage Organoid Formation: A Phenotypic, Molecular, and Functional Characterization

**DOI:** 10.3390/cells14181423

**Published:** 2025-09-11

**Authors:** Cristian Mera Azain, Astrid Natalia Santamaría Durán, Tatiana Camila Castañeda, Luis Fernando Useche, Efraín Leal Garcia, Jaime Mariño Valero, Rodrigo Jaramillo Quintero, Luis Fernando Jaramillo, Jorge Andrés Franco, Rubiela Castañeda Salazar, Juan Carlos Ulloa, Ivonne Gutiérrez Rojas, Rodrigo Somoza Palacios, Claudia Cuervo Patiño, Viviana Marcela Rodríguez-Pardo

**Affiliations:** 1Grupo Inmunobiología y Biología Celular, Departamento de Microbiología, Facultad de Ciencias, Pontificia Universidad Javeriana, Bogotá 110231, Colombia; mera.cristian@javeriana.edu.co (C.M.A.); astrid.santamaria@javeriana.edu.co (A.N.S.D.); tcastaneda@javeriana.edu.co (T.C.C.); 2Departamento de Ortopedia y Traumatología, Hospital Universitario San Ignacio, Facultad de Medicina, Pontificia Universidad Javeriana, Bogotá 110231, Colombia; useche.l@javeriana.edu.co (L.F.U.); eleal@husi.org.co (E.L.G.); j.marinov@javeriana.edu.co (J.M.V.); rjquintero@husi.org.co (R.J.Q.); 3Departamento de Patología, Hospital Universitario San Ignacio, Facultad de Medicina, Pontificia Universidad Javeriana, Bogotá 110231, Colombia; luis.jaramillo@javeriana.edu.co; 4Departamento de Morfología, Facultad de Medicina, Pontificia Universidad Javeriana, Bogotá 110231, Colombia; jafrancoz@javeriana.edu.co; 5Unidad de Investigaciones Agropecuarias (UNIDIA), Departamento de Microbiología, Pontificia Universidad Javeriana, Bogotá 110231, Colombia; castaneda.r@javeriana.edu.co; 6Grupo de Enfermedades Infecciosas, Departamento de Microbiología, Pontificia Universidad Javeriana, Bogotá 110231, Colombia; julloa@javeriana.edu.co (J.C.U.); claudia.cuervo@javeriana.edu.co (C.C.P.); 7Grupo de Biotecnología Ambiental e Industrial (GBAI), Departamento de Microbiología, Pontificia Universidad Javeriana, Bogotá 110231, Colombia; ivonne.gutierrez@javeriana.edu.co; 8Center for the Multimodal Evaluation of Structural Tissues, Department of Biology, Case Western Reserve University, Cleveland, OH 44106, USA; ras286@case.edu

**Keywords:** cartilage organoids, chondrogenesis, mesenchymal stem cells, skeletal stem cells, physioxia

## Abstract

Articular cartilage regeneration remains a major challenge due to its limited self-repair capacity. Bone marrow-derived skeletal stem cells (SSCs) and mesenchymal stem cells (MSCs) are promising candidates for cartilage engineering, although they differ in their chondrogenic potential. This study explored whether co-culturing SSCs and MSCs in three-dimensional (3D) organoid systems under cartilage physioxia (5% O_2_) and chondrogenic induction could improve cartilage tissue formation. SSCs, MSCs, and SSC–MSC co-cultures were characterized for morphology, phenotype, and differentiation capacity. Organoids were generated and cultured for 10 days, followed by analysis of morphology, viability, gene expression (*SOX9*, *RUNX2*, *ACAN*, *COL2A1*, *COL10A1*, *PRG4*, and *PDPN*), chondrocyte-associated antigens (CD44, CD105, CD146, and PDPN), and cartilage ECM proteins (aggrecan, collagen types I, II, and X, and PRG4). SSCs showed robust chondrogenic and osteogenic potential, while MSCs exhibited a balanced multipotency. Co-culture-derived organoids enhanced chondrogenesis and reduced adipogenesis, with higher expression of cartilage-specific ECM and lower hypertrophic marker levels. These findings highlight the functional synergy between SSCs and MSCs in co-culture, promoting the formation of stable, cartilage-like structures under physioxia. The approach offers a promising strategy for generating preclinical models and advancing regenerative therapies for hyaline cartilage repair.

## 1. Introduction

Articular hyaline cartilage is an avascular and aneural tissue that covers the bone surfaces of synovial joints, providing resistance to compression and reducing friction during movement. It consists of a predominant cell population, chondrocytes, embedded in an extracellular matrix (ECM) rich in type II collagen, aggrecan, and other structural proteins essential for its biomechanical function [1,2,3]. However, due to its limited endogenous regenerative capacity, articular cartilage injuries tend to progress toward degenerative diseases such as osteoarthritis, affecting the quality of life of millions of people worldwide [4,5]. Currently available treatments, such as microfracture, autologous chondrocyte implantation (ACI), and allografts, provide symptomatic relief but fail to achieve complete cartilage tissue regeneration, thereby limiting their long-term efficacy [6].

In this context, tissue bioengineering has explored the development of 3D cultures or cartilage organoids derived from stem cells as a promising strategy for articular cartilage repair. Cartilage organoids are three-dimensional structures derived from stem cells that mimic the cellular organization and ECM composition of native cartilage, providing a physiologically relevant model for its study and clinical application [7,8]. Nevertheless, questions remain regarding the optimal stem cell type for generating organoids that closely resemble native cartilage and possess in vivo regenerative potential [9]. In this regard, although both mesenchymal stem cells (MSCs) [10] and skeletal stem cells (SSCs) [11] have demonstrated chondrogenic potential. MSCs have been shown to generate hypertrophic chondrocytes or fibrocartilage-producing cells, which are more associated with inflammation and tissue damage than with the formation of native hyaline cartilage [12,13,14]. This observation raises questions about their suitability as a cellular source for cartilage regeneration strategies. Moreover, it remains unclear whether the interaction between these two chondrogenic populations could yield optimized organoids for tissue repair, particularly in the context of the stem cell niche hypothesis [15,16]. According to this hypothesis, MSCs primarily function as support cells for a more restricted and tissue-specific stem cell population: in this case, SSCs, which would be directly responsible for cartilage regeneration. Therefore, further characterization of the functional roles of each cell type within the organoid microenvironment is necessary, along with additional studies to determine whether MSC–SSC co-culture enhances the structural stability of the resulting organoids and reduces the risk of hypertrophy and fibrosis in cartilage tissue.

Another critical factor in the formation of functional cartilage is oxygen tension. It has been demonstrated that physiological oxygen levels in articular cartilage (1–5% O_2_) promote chondrogenic differentiation and ECM production characteristic of hyaline cartilage, whereas in vitro normoxia (21% O_2_) induces a hypertrophic phenotype and the production of type X collagen, which is associated with endochondral ossification [17,18,19]. However, there is limited evidence regarding the effect of cartilage physiological oxygen levels or cartilage physioxia on cartilage organoid generation.

In cartilage organoid development, in addition to identifying the optimal stem cell populations for their generation and defining appropriate physical conditions, such as oxygen levels, ECM protein production within these structures is a key factor in their potential clinical application. In cartilage organoid models, ECM characterization has primarily focused on analyzing classical structural proteins, such as type II collagen and aggrecan, due to their essential roles in cartilage formation and functionality. However, other cartilage-associated proteins with key functions during specific developmental stages and in determining tissue quality have been scarcely explored in these models. Among them, podoplanin (PDPN), involved in prenatal chondrogenesis, and proteoglycan 4 (PRG4), associated with joint lubrication and considered to be a potential prognostic marker of success in cartilage regeneration strategies, stand out [20,21]. PDPN is a transmembrane glycoprotein expressed in chondrocytes during embryonic development and in the growth plate, playing a role in cellular organization and cartilage stability [22,23]. Its expression has been associated with cytoskeletal regulation and cell–ECM communication [24], although its function in cartilage organoids generated in vitro remains unclear. On the other hand, PRG4, also known as lubricin, is a key protein for cartilage lubrication and the prevention of joint friction [25]. Its expression has been linked to optimal tribological properties following cartilage regeneration processes [26]. Moreover, PRG4 expression increases after the successful implantation of human embryonic stem cell (hESC)-derived chondrocytes [27] and induced pluripotent stem cell (iPSC)-derived cartilage organoids [20], suggesting its potential as a biomarker for successful organoid transplantation in cartilage regeneration therapies. Despite advancements in organoid development, it remains undetermined whether PDPN and PRG4 expression in these structures is constitutive or dependent on specific conditions, such as physiological cartilage physioxia. Furthermore, it is unknown whether their expression correlates with the production of chondrocytes or ECM proteins characteristic of hyaline cartilage.

Since cartilage regeneration represents a significant clinical challenge, and current treatments fail to fully restore its structure and function, it is essential to deepen our understanding of cartilage organoids and identify key biomarkers associated with their development and quality. In this context, the present study contributes to this knowledge by generating and characterizing the structural and functional properties of cartilage organoids derived from mesenchymal and skeletal stem cells, cultured under cartilage physioxia conditions that replicate the articular cartilage microenvironment. Additionally, we evaluated classical ECM proteins, such as type II collagen and aggrecan, as well as other proteins relevant to chondrogenesis or tissue quality indicators, including podoplanin (PDPN) and proteoglycan 4 (PRG4), which have been scarcely studied in cartilage organoids.

## 2. Materials and Methods

### 2.1. Donor Selection and Bone Marrow Sample Collection

Bone marrow (BM) samples were obtained from patients undergoing prosthetic hip replacement at the Department of Orthopedics and Traumatology, Hospital Universitario San Ignacio (Bogotá, Colombia), who voluntarily agreed to participate in this study. Written informed consent was obtained in accordance with approvals from the Research and Ethics Committees of the Faculty of Sciences at Pontificia Universidad Javeriana (Acta No. 10, 17 June 2021) and Hospital Universitario San Ignacio (Acta No. 11, 24 June 2021). Samples were collected by orthopedic surgeons during surgery and subsequently transported to the Faculty of Sciences at Pontificia Universidad Javeriana for processing and analysis. Donor selection was conducted according to previously published inclusion criteria [28,29]. Eligible patients were those over 18 years of age, with no prior diagnosis of hematopoietic system neoplasms; seronegative for hepatitis B virus (HBV), hepatitis C virus (HCV), and human immunodeficiency virus (HIV); and not receiving immunosuppressive treatments.

### 2.2. Isolation and Culture of Skeletal Stem Cells (SSCs) and Mesenchymal Stem Cells (MSCs)

Mononuclear cells (MNCs) were isolated from BM samples using a density gradient with Histopaque 1077 (d = 1.077 g/cm^3^, Sigma-Aldrich^®^, (Merck KGaA), Burlington, MA, USA). Specific cell populations were then separated to obtain MSCs and SSCs. For MSC isolation, MNCs were directly cultured at a density of 160 × 10^3^ cells/cm^2^ in IMDM medium (Gibco, Thermo Scientific^®^, Waltham, MA, USA) supplemented with 10% fetal bovine serum (FBS, Gibco, Thermo Scientific^®^, Waltham, MA, USA), 1% sodium pyruvate (Sigma-Aldrich^®^, (Merck KGaA), Burlington, MA, USA), 1% non-essential amino acids (Gibco, Thermo Scientific^®^, Waltham, MA, USA), and 1% penicillin/streptomycin (Eurobio^®^, Eurobio Scientific, Les Ulis, France). Cultures were maintained at 37 °C and 5% CO_2_ for 72 h, after which non-adherent cells were removed. Adherent cells with fibroblast-like morphology were cultured until reaching 70% confluence [30,31,32]. For SSC isolation, a negative immunomagnetic separation was performed from 10 × 10^6^ MNCs using anti-CD45 MicroBeads (Miltenyi Biotec^®^, Bergisch Gladbach, Germany), followed by a second negative immunomagnetic separation using anti-CD146-PE and anti-PE MicroBeads (Miltenyi Biotec^®^, Bergisch Gladbach, Germany), yielding CD45-/CD146- cells. A subsequent positive immunomagnetic selection was performed using anti-CD73-PE, anti-CD164-APC, anti-PE, and anti-APC MicroBeads (Miltenyi Biotec^®^, Bergisch Gladbach, Germany) to obtain CD45-/CD146-/CD73+/CD164+ cells. These cells were cultured at a density of 2 × 10^4^/cm^2^ in MEM-alpha medium (Gibco, Thermo Scientific^®^, Waltham, MA, USA) supplemented with 10% human platelet lysate (STEMCELL Technologies^®^, STEMCELL Technologies Inc., Vancouver, BC, Canada), 1% sodium pyruvate (Sigma-Aldrich^®^, (Merck KGaA), Burlington, MA, USA), and 1% penicillin/streptomycin (Eurobio^®^, Eurobio Scientific, Les Ulis, France). Cultures were maintained at 37 °C and 5% CO_2_ for 72 h, after which non-adherent cells were removed. Adherent cells with fibroblast-like morphology were cultured until reaching over 70% confluence [33,34]. For both MSCs and SSCs, three passages of the primary culture were performed, followed by individual morphological and phenotypic characterization. Subsequently, MSCs and SSCs from each donor were pooled to create a homogeneous population for each cell type, which was then subjected to morphological, phenotypic, and functional characterization. Pooling of each cell population and their co-culture was conducted to reduce donor-dependent biological variability in the isolated cells, as previously demonstrated [35,36].

### 2.3. Morphological, Phenotypic, and Functional Characterization of SSC, MSC, and SSC-MSC Pools

Three cell pools were generated from the characterized primary cultures: SSCs, MSCs, and an SSC-MSC co-culture at a 1:1 ratio. Morphological, phenotypic, and functional characterization of the pools was performed in accordance with ISO 24651 standards [37] for MSCs and experimental background for SSCs [31,34]. Morphological characterization was conducted using inverted microscopy with an OLYMPUS CKX3 microscope. For a more detailed morphological evaluation, cells were subjected to cytocentrifugation in an Aerospray^®^ Hematology Pro system. Cells were then stained with Wright’s stain (Sigma-Aldrich^®^, (Merck KGaA), Burlington, MA, USA) and evaluated under a conventional optical microscope. The phenotype of the cell populations was determined using qRT-PCR to assess the expression of pluripotency-associated genes [38,39,40,41] and spectral flow cytometry (Aurora, Cytek^®^ Biosciences Inc., Fremont, CA, USA). Flow cytometry data were analyzed using FlowJo v10.10.0 to establish the antigenic profile reported for each cell population [11,31]. The gating strategy included doublet and debris exclusion, selection of highly viable cells, and generation of fluorescence intensity histograms to determine the median fluorescence intensity (MFI) of each antigen.

Functional characterization of the three cell pools (SSCs, MSCs, and SSC-MSC co-culture) was performed through colony formation and multipotent differentiation assays. Colony formation assays involved culturing 0.5 × 10^2^ cells from each pool in their respective culture medium. After 14 days at 37 °C and 5% CO_2_, colonies were fixed with 4% paraformaldehyde (Sigma-Aldrich^®^, (Merck KGaA), Burlington, MA, USA) and stained with 3% crystal violet (Sigma-Aldrich^®^ (Merck KGaA), Burlington, MA, USA) for counting. Multipotent differentiation capacity was assessed by culturing cells in lineage-specific induction media: osteogenesis (StemPro^®^ Osteogenesis Differentiation Kit, Gibco, Thermo Scientific^®^, Waltham, MA, USA), adipogenesis (StemPro^®^ Adipogenesis Differentiation Kit, Gibco, Thermo Scientific^®^, Waltham, MA, USA), and chondrogenesis (StemPro^®^ Chondrogenesis Differentiation Kit, Gibco, Thermo Scientific^®^, Waltham, MA, USA). Osteogenesis was confirmed by Von Kossa staining (Sigma-Aldrich^®^, (Merck KGaA), Burlington, MA, USA) for calcium deposits, adipogenesis by Oil Red O staining (Sigma-Aldrich^®^, (Merck KGaA), Burlington, MA, USA) for intracellular lipid vacuoles, and chondrogenesis by Alcian Blue staining for glycosaminoglycans (GAG) (Santa Cruz Biotechnology^®^, Dallas, TX, USA). Multipotency was further evaluated through qRT-PCR expression analysis of lineage-specific transcription factors: *RUNX2* (osteogenesis), *PPARγ* (adipogenesis), and *SOX9* (chondrogenesis) (Table 1) [42].

### 2.4. Maintenance of Control Cell Lines: NHAC-kn and HPdLF

To compare the phenotypic and functional characteristics of the obtained pools and, subsequently, the characteristics of the developed organoids, control cell lines were used, including cartilage-constituting cells (articular chondrocytes) and cells not associated with articular hyaline cartilage (gingival fibroblasts). The Human Knee Articular Chondrocytes (NHAC-kn) CC-2550 cell line (Lonza^®^, Lonza Group Ltd., Basel, Switzerland) was cultured in CGM™ Chondrocyte Growth Medium BulletKit™ supplemented with 1% penicillin/streptomycin (Gibco, Thermo Scientific^®^, Waltham, MA, USA). The Human Periodontal Ligament Fibroblasts (HPdLF) CC-7049 cell line (Lonza^®^, Lonza Group Ltd., Basel, Switzerland) was cultured in RPMI (STEMCELL Technologies^®^, STEMCELL Technologies Inc., Vancouver, BC, Canada) supplemented with 1% penicillin/streptomycin (Eurobio^®^, Eurobio Scientific, Les Ulis, France). These cells were provided by the Centro de Investigaciones Odontológicas (CIO) of Pontificia Universidad Javeriana.

### 2.5. Cartilage Organoid Generation from SSCs, MSCs, and SSC-MSC Co-Culture

Organoids were generated from the three previously described cell pools, which included skeletal stem cells (SSCs), mesenchymal stem cells (MSCs), and a 1:1 co-culture pool of SSCs and MSCs. Each of these cell conditions was seeded into ultralow-attachment 96-well plates (Corning^®^, Corning Inc., Corning, NY, USA) at a density of 1 × 10^5^ cells per well, using an appropriate volume of culture medium. Two culture conditions were established: without chondrogenic induction medium (−IM), using DMEM-F12 (Gibco, Thermo Scientific^®^, Waltham, MA, USA) supplemented with 10% KnockOut™ serum (Gibco, Thermo Scientific^®^, Waltham, MA, USA) and 1% penicillin/streptomycin (Eurobio^®^, Eurobio Scientific, Les Ulis, France); and with chondrogenic induction medium (+IM), using the StemPro^®^ Chondrogenesis Differentiation Kit (Gibco, Thermo Scientific^®^, Waltham, MA, USA) under the same supplementation conditions. After seeding each pool, the plates were centrifuged at 500× *g* for 15 min to promote cell aggregation and then incubated under physioxic conditions (5% O_2_, 5% CO_2_, 37 °C) for a period of 10 days. Organoid formation was based on spontaneous cell self-aggregation without the use of scaffolds, facilitated by the low-adhesion surface of the culture plates and the physioxic environment, which closely mimics the native cartilage microenvironment. Simultaneously, control cell lines (NHAC-kn and HPdLF) were cultured under the same conditions, starting with 1 × 10^3^ HPdLF cells and 4 × 10^3^ NHAC-kn cells. All of the experiments described below were performed in biological triplicates to ensure reproducibility and to allow for appropriate statistical analysis of the results.

### 2.6. Evaluation of Cartilage Organoid Morphology, Viability, and Physioxia Condition Validation

The evaluation of morphology, viability, and validation of cartilage physioxia (5% O_2_) conditions in the organoids was conducted on days 1, 5, and 10, with data acquisition performed using the Cytation 5 Cell Imaging Multimode Reader (BioTek^®^, BioTek Instruments, Inc., Winooski, VT, USA). Morphological evaluation was carried out by measuring the diameters of the 3D structures in the culture plate using the Line Tool in the Gen5 v. 3.13 software on the multifunctional imaging system. Viability assessment of organoids and control cell lines was performed on days 1, 5, and 10 using the LIVE/DEAD Cell Imaging Kit (Invitrogen, Thermo Scientific^®^, Waltham, MA, USA), following the manufacturer’s protocol. Image acquisition was conducted using the multifunctional imaging system, and data analysis was performed with ImageJ software v2.140/1.54F. Validation of physioxia conditions in 3D structures was conducted using the BioTracker™ 520 Green Hypoxia Dye (Sigma-Aldrich^®^, (Merck KGaA), Burlington, MA, USA) [43], and data analysis was performed using ImageJ software v2.140/1.54F.

### 2.7. Evaluation of Gene and Protein Expression Associated with Chondrogenesis in Cartilage Organoids Under Physioxia

After 10 days of culture, total RNA was extracted from the organoids using the RNeasy Plus Mini Kit (Qiagen^®^, QIAGEN GmbH, Hilden, Germany), following the manufacturer’s protocol, and qRT-PCR was performed according to the manufacturer’s instructions. Two-step quantitative real-time PCR (qRT-PCR) was performed. cDNA was synthesized from 1 ng/µL of total RNA using a QuantiTect Reverse Transcription Kit (Qiagen^®^, QIAGEN GmbH, Hilden, Germany). Quantitative PCR (qPCR) was carried out using PowerUpTM SYBRTM Green Master Mix” (Biosystems^®^, BioSystems S.A., Barcelona, Spain). Each target and housekeeping gene was analyzed in three technical replicates [44]. The expression levels of *SOX9*, *ACAN*, *PDPN*, *PGR4*, *RUNX2*, and *COL10* mRNA were determined. Expression levels were normalized using the housekeeping genes *B2M*, *GAPDH*, and *ACTB* (Table 1), following the ΔΔCt method [45]. Protein expression associated with chondrocyte generation in organoids was assessed through enzymatic digestion (1% trypsin, Eurobio^®^, Eurobio Scientific, Les Ulis, France) and mechanical dissociation by repeated pipetting. The resulting cells were transferred to flow cytometry tubes with a 35 μm cell strainer (Falcon™, Corning Inc., Corning, NY, USA), incubated with Zombie Aqua Fixable Viability dye (Biolegend^®^, BioLegend Inc., San Diego, CA, USA), and subsequently stained with the following monoclonal antibodies: CD44-PerCy7 (Clone G44-26, BD Pharmingen^®^, BD Biosciences, San Jose, CA, USA), CD105 BV421 (Clone SN6h, Biolegend^®^, BioLegend Inc., San Diego, CA, USA), CD146 BV605 (Clone P1H12, Biolegend^®^, BioLegend Inc., San Diego, CA, USA), and PDPN APC (Clone NC-08, Biolegend^®^, BioLegend Inc., San Diego, CA, USA). Data acquisition was performed using the Cytek Aurora^®^ spectral flow cytometer, and analysis was conducted with FlowJo v10.10.0 software. The immunophenotype was represented using t-SNE dimensionality reduction algorithms and heatmaps based on the mean fluorescence intensity of each antigen in viable cells [46]. Appendix A shows the primers used for the qRT-PCR assays.

### 2.8. Determination of Cartilage Extracellular Matrix Proteins in Organoids

After 10 days of culture, organoids were transferred to Eppendorf tubes with 0.5% gelatin (Sigma-Aldrich^®^) for transportation to the Pathology Department at the Hospital Universitario San Ignacio. The 3D structures were then embedded in CryoMatrix (Feather Health^®^, Feather Health, 24 Broadcast Drive, Charlotte, NC, USA) and sectioned in a KD-3000 cryostat microtome at −20 °C to obtain 5 µm thick slices. Sections were subsequently transferred to 98% ethanol for one minute and then incubated at 80 °C. For immunohistochemistry (IHC), the Autostainer Link 48 automated system was used with the following antibodies: aggrecan (Clone BC-3, Novus^®^, Novus Biologicals, Centennial, CO, USA), collagen I alpha 1 (Clone COL-1, Novus^®^, Novus Biologicals, Centennial, CO, USA), collagen II (Clone 5B2.5, Novus^®^, Novus Biologicals, Centennial, CO, USA), collagen X alpha 1 (Clone SR3302, Novus^®^, Novus Biologicals, Centennial, CO, USA), and PRG4 (NBP1-19048, Novus^®^, Novus Biologicals, Centennial, CO, USA). IHC results were analyzed by double-blind microscopic evaluation performed by pathologists from the Pathology Department at the Hospital Universitario San Ignacio. The results were then tabulated and assigned scores as follows: 0 (negative), 1 (mildly positive), 2 (moderately positive), or 3 (intensely positive). Negative controls consisted of adipose tissue sections, while positive controls included sections of fetal trachea, skin, and intervertebral disc.

### 2.9. Statistical Analysis

The Shapiro–Wilk test was used to assess data normality. Based on these results, the Mann–Whitney U test and Kruskal–Wallis test, followed by Dunn’s multiple comparison test, were applied for non-parametric distributions. For parametric distributions, one-way ANOVA followed by Tukey’s multiple comparison test was performed. Statistical significance was set at *p* ≤ 0.05. Statistical analyses were conducted using GraphPad Prism v10.1.2.324.

## 3. Results

### 3.1. Characteristics of Donors and Collected Human Bone Marrow Samples

With the support of the medical team from the Department of Orthopedics and Traumatology at the Hospital Universitario San Ignacio (Bogotá, Colombia), six bone marrow (BM) samples were collected for the isolation of skeletal stem cells (SSCs) and six samples for the isolation of mesenchymal stem cells (MSCs). The SSC isolation samples were obtained from voluntary donors aged 55 to 73 years, of whom 66.6% were male and 33.5% were female, with an average collected bone marrow volume of 68.5 mL. The MSC isolation samples were obtained from donors aged 60 to 84 years, with a distribution of 66.5% male and 33.5% female, and an average collected sample volume of 82.5 mL (Table 1).

### 3.2. Phenotypic and Functional Characterization of SSC Pools

The SSC pool yielded fibroblast-like cells with an average length of 128.4 μm and the presence of perinuclear vacuoles (Figure 1A). Morphological analysis using cytospin revealed cells with an average diameter of 29.8 μm, loose chromatin, 2 to 3 nucleoli, basophilic cytoplasm, cytoplasmic vacuoles, and a nucleus-to-cytoplasm ratio of 4:6 (Figure 1B). Immunophenotypic characterization confirmed that SSCs lacked hematopoietic (CD34 and CD45) and endothelial (CD31) antigens; 99% of the population expressed typical SSC markers—CD73, CD105, and CD164—while CD146 was absent. Additionally, 69.1% expressed PDPN (Figure 1C). In colony-forming assays, the SSC pool generated between 17 and 22 colony-forming units (CFUs), with diameters ranging from 0.4 to 0.7 mm, from 50 seeded cells (Figure 2A,B). Chondrogenic Differentiation: Cells exposed to induction medium exhibited high glycosaminoglycan (GAG) production, as evidenced by Alcian Blue staining (Figure 2C), along with *SOX9* mRNA expression higher than that of control cells (Figure 2D).

Osteogenic Differentiation: Von Kossa staining revealed calcium salt deposits in induced cells (Figure 2E). Although *RUNX2* expression did not show significant differences (*p* = 0.1287) (Figure 2F), it demonstrated an upward trend, reaching six times the expression level of control cells. Adipogenic Differentiation: No lipid vacuoles or morphological changes were observed via Oil Red O staining (Figure 2G), suggesting that SSCs selectively differentiate into osteochondral lineages. However, *PPARγ* expression increased significantly (*p* = 0.0082) (Figure 2H), possibly indicating its involvement in other cellular processes beyond adipogenic differentiation [47]. The isolated SSCs correspond to a previously described population in the literature, characterized by fibroblast-like morphology, expression of CD73, CD105, CD164, and PDPN, and the absence of hematopoietic and endothelial markers. Their clonogenic capacity, preference for osteochondral differentiation, and limited adipogenic potential align with the functional profile of human bone marrow-derived skeletal stem cells [11,21,33,34,48].

### 3.3. Phenotypic and Functional Characterization of MSC Pools

The MSC pool displayed fibroblast-like morphology, with an average cell length of 316.1 μm (Figure 3A). Morphological analysis via cytospin revealed loosely condensed chromatin, 1 to 2 nucleoli, basophilic cytoplasm, and a nucleus-to-cytoplasm ratio of 4:6 (Figure 3B). Immunophenotypic analysis via spectral flow cytometry (Cytek Aurora^®^) confirmed the absence of hematopoietic (CD34 and CD45) and endothelial (CD31) antigens; 99% of the population expressed CD73, CD105, and CD164, while 84.7% expressed CD146 and 75% expressed PDPN (Figure 3C). In colony-forming assays, the MSC pool generated between 12 and 17 CFUs, with diameters ranging from 2 to 3.5 mm (Figure 4A,B). Chondrogenic Differentiation: High GAG production was evidenced by Alcian Blue staining (Figure 4C), along with an 18-fold increase in *SOX9* expression compared to control cells (Figure 4D). Osteogenic Differentiation: Von Kossa staining revealed calcium deposits (Figure 4E), but *RUNX2* expression showed no significant differences (*p* = 0.3744) and even decreased slightly in induced cells (Figure 4F). Adipogenic Differentiation: Lipid vacuoles were observed via Oil Red O staining (Figure 4G), along with a 20-fold increase in *PPARγ* expression compared to non-induced cells (Figure 4H). These phenotypic and functional characteristics align with the previous literature and ISO 24651 standards [28,30,31,49].

### 3.4. Phenotypic and Functional Characterization of SSC-MSC Co-Culture Pools

The combined SSC-MSC pool was co-cultured, where fibroblast-like morphology was observed (Figure 5A). Cytospin analysis revealed loosely condensed chromatin, 1 to 2 nucleoli, basophilic cytoplasm, cytoplasmic vacuoles, and a nucleus-to-cytoplasm ratio of 4:6 (Figure 5B). Spectral flow cytometry (Cytek Aurora^®^) analysis confirmed the absence of hematopoietic (CD34 and CD45) and endothelial (CD31) antigens; 99% of the population expressed CD73, CD105, and CD164. A heterogeneous CD146 expression pattern was observed, with 55.2% negative cells and 44.7% positive cells (Figure 5C). In functional assays, the SSC-MSC pool generated between 15 and 22 CFUs from 50 seeded cells (Figure 6A,B). Chondrogenic Differentiation: The interaction between SSCs and MSCs enhanced GAG production, as evidenced by Alcian Blue staining (Figure 6C), and led to a remarkable increase in *SOX9* expression, with levels 40 times higher than the SSC pool and 50 times higher than the MSC pool (Figure 6D). Osteogenic Differentiation: Positive mineralization was observed via Von Kossa staining (Figure 6E), and *RUNX2* expression remained unchanged in the co-culture (Figure 6F). Adipogenic Differentiation: Lipid vacuoles were detected, but adipogenic potential was reduced, as indicated by fewer cells with vacuoles (Figure 6G) and decreased *PPARγ* expression compared to the individual SSC and MSC pools (Figure 6H). No prior studies have explored the synergistic effects of SSCs and MSCs in cartilage organoid formation. While phenotypic characterization was consistent with the combination of both cell populations, functional evaluation revealed a significant increase in *SOX9* expression and a sharp decrease in *PPARγ*, suggesting that SSC-MSC interaction may enhance chondrogenic potential, promoting commitment to the cartilage lineage.

As part of the characterization of the three cell pools, the expression of pluripotency genes, including *NANOG*, *OCT3/4*, and *TRA1-81*, was also assessed. All cell pools exhibited expression of these genes, which may be associated with their stem cell characteristics. However, the highest expression levels were observed in SSCs (Figure 7A–C), which could be related to the isolation strategy used for this cell population. Nevertheless, functional assays are required to determine the pluripotent capacity of these cells. In relation to the comparison of multipotent differentiation potential based on the expression of lineage-specific genes—*SOX9* (chondrogenesis), *RUNX2* (osteogenesis), and *PPARγ* (adipogenesis)—among pools derived from SSCs, MSCs, and SSC–MSC co-cultures, it is noteworthy that the highest *SOX9* expression, coupled with the lowest *PPARγ* expression, was observed in cartilage organoids derived from the SSC–MSC co-culture. This finding suggests that the interaction between these two stem cell populations may enhance chondrogenic commitment while suppressing adipogenic differentiation pathways (Figure 7D–F).

### 3.5. Phenotypic Characterization of Control Populations: NHAC-kn and HPdLF

In culture, the NHAC-kn and HPdLF control cell lines exhibited fibroblast-like morphology. NHAC-kn Cytospin Analysis: Condensed chromatin, absence of nucleoli, basophilic cytoplasm, and a nucleus-to-cytoplasm ratio of 3:7 (Figure 8A,B). HPdLF Cytospin Analysis: Loosely condensed chromatin, 1 to 2 nucleoli, slightly basophilic cytoplasm, cytoplasmic vacuoles, and granules, with a nucleus-to-cytoplasm ratio of 6:4 (Figure 8C,D). Flow cytometry analysis distinguished two different populations, where HPdLF fibroblasts did not express the evaluated antigens, while NHAC-kn chondrocytes expressed CD73 (99%), CD105 (52%), CD164 (96%), and PDPN (44%) (Figure 8E).

### 3.6. Formation and Characterization of Cartilage Organoids Derived from Stem Cells

The morphology, viability, and validation of physioxia in organoids were assessed on days 1, 5, and 10 of culture. Over time, a decrease in organoid diameter was observed, possibly due to aggregation and compaction processes characteristic of 3D cultures (Figure 9A). This reduction was less pronounced in organoids treated with induction medium (+IM), suggesting a higher production of extracellular matrix (ECM) proteins. By day 10, SSC organoids (−IM) exhibited a 31.5% reduction, whereas SSC organoids (+IM) showed a smaller decrease of 27.8%. A similar trend was observed in MSC organoids, with a 36.7% diameter reduction in the −IM condition and 22.8% in the +IM condition. In SSC-MSC co-culture organoids, the reduction was 27.5% without induction and 14.1% with induction medium (Figure 9C). Cell viability remained above 90% in all evaluated conditions under 5% O_2_ (Figure 9B,D).

Organoids were generated under cartilage physioxia conditions (5% O_2_) for 10 days, simulating the native articular cartilage environment. Physioxia validation was performed using a fluorescent probe sensitive to low oxygen levels [43]. Both control and experimental conditions showed increased fluorescence when comparing cultures in normoxia (21% O_2_) vs. physioxia (5% O_2_) (Figure 10). The NHAC-kn control cell line exhibited a 76-fold increase in fluorescence on day 5 and an 82-fold increase on day 10, while HPdLF fibroblasts showed 20-fold and 27-fold increases, respectively (Figure 10A,B). Notably, SSC-derived organoids and SSC-MSC co-culture organoids exhibited fluorescence increases of up to 150-fold, while MSC-derived organoids reached up to 130-fold increases, when comparing physioxia versus normoxia cultures (Figure 10C,D). The evaluation of morphology, viability, and physioxia validation demonstrated that organoids generated under 5% O_2_ maintained high cell viability and underwent progressive compaction, possibly associated with ECM production under chondrogenic induction medium (+IM). Furthermore, fluorescent probe-based physioxia validation confirmed that all organoids and control lines adequately responded to reduced oxygen levels, with significant fluorescence signal increases, particularly in SSC and SSC-MSC organoids, which exhibited the highest increments. These findings suggest that the established culture conditions successfully replicate an appropriate microenvironment for cartilage organoid formation, consistent with the physiological environment of articular cartilage.

### 3.7. Evaluation of Cartilage-Associated Gene Expression in Organoids Under Physiological Physioxia

The analysis of cartilage-associated gene expression in organoids is crucial to assessing their ability to mimic the molecular characteristics of articular cartilage, as well as their potential to reproduce key aspects of cartilage development and homeostasis. To this end, the transcriptional expression of key chondrogenesis and cartilage function genes was analyzed, including *SOX9*, *ACAN*, *PDPN*, *PRG4*, *RUNX2*, and *COL10*, comparing organoids cultured with (+IM) and without (−IM) chondrogenic induction medium, as well as against the NHAC-kn control articular chondrocyte line. SSC and MSC organoids treated with +IM exhibited a more than 10-fold increase in *SOX9* expression compared to organoids cultured without IM, similar to the trend observed in the SSC-MSC co-culture organoids (Figure 11A). Although ACAN expression did not show statistically significant differences between organoids with and without IM, a clear upward trend was observed under IM conditions, with levels five times higher than in non-induced organoids (Figure 11B). *PDPN* gene expression showed an increasing trend in IM-treated organoids, although it did not reach statistical significance compared to −IM conditions. However, among organoids, the SSC-MSC co-culture-derived organoids exhibited the highest *PDPN* expression levels relative to their −IM counterparts (Figure 11C). PRG4, a key protein for joint lubrication and ECM maintenance, exhibited significantly higher expression in IM-treated organoids compared to their non-treated counterparts (Figure 11D). *RUNX2* and COL10A1 expression followed a similar pattern, with higher expression in organoids exposed to IM (Figure 11E,F).

Organoids demonstrated a gene expression profile similar to that of articular chondrocytes, particularly in the expression of *SOX9*, *ACAN*, *PDPN*, and *RUNX2*, with no significant differences compared to the NHAC-kn control line. This result is highly promising, as it suggests that the organoids successfully replicate key characteristics of the control chondrocyte line, reinforcing their potential for cartilage regeneration applications. A particularly relevant finding was the expression of PRG4 in IM-treated organoids, albeit at lower levels than in chondrocytes. Since PRG4 expression is particularly modulated by the biomechanical forces of cartilage [50], its detection in organoids despite not being exposed to such conditions is an encouraging result. This suggests that organoids may be acquiring tissue-quality characteristics, and that PRG4 expression could serve as a prognostic indicator of successful transplantation and functional integration in a physiological environment.

### 3.8. Generation of Chondrocytes from Cartilage Organoids Derived from SSCs, MSCs, and SSC–MSC Co-Cultures

To evaluate the antigenic profile associated with chondrocytes in organoids cultured with chondrogenic induction medium (+IM), the expression of chondrocyte-associated antigens was analyzed: CD44, CD105, CD146, and PDPN [22,51]. Additionally, the expression levels of these antigens were compared with those of control cell lines of chondrocytes and fibroblasts (Figure 12). The chondrocyte control line exhibited high expression of CD44, CD105, CD146, and PDPN, whereas fibroblasts showed low CD44 expression and absence of the other evaluated antigens. SSC-derived organoids exhibited lower expression of CD44, CD105, and CD146 compared to chondrocytes and the other conditions (Figure 12A); however, PDPN expression was higher compared to MSC-derived organoids and SSC-MSC co-culture organoids (Figure 12B,C). MSC-derived organoids expressed CD44, CD105, and CD146, but they had the lowest PDPN expression among the three types of organoids (Figure 12B). Interestingly, SSC-MSC co-culture organoids expressed all of the evaluated antigens, albeit at lower levels compared to chondrocytes (Figure 12C).

This evaluation identified PDPN as a potentially relevant marker, particularly in SSC-derived organoids and SSC-MSC co-culture organoids, which exhibited significantly higher PDPN expression compared to MSC-derived organoids. The high PDPN expression suggests a possible association with an immature chondrogenic phenotype, characteristic of early differentiation stages toward hyaline cartilage, as PDPN expression has been demonstrated during prenatal chondrogenesis [23,52]. In contrast, the low PDPN expression observed in MSC-derived organoids may reflect greater plasticity toward other cell lineages, potentially compromising the chondrogenic stability of these organoids. The antigenic profile evaluation in organoids provided insight into chondrocyte generation within these structures. However, these results are not conclusive, as confirming chondrocyte generation or maturation status requires cell isolation and further phenotypic and functional assays to validate their identity. Nonetheless, these findings are particularly interesting in SSC-MSC co-culture organoids (Figure 12C), where the expression of all evaluated antigens suggests that cellular interaction may enhance chondrogenesis.

### 3.9. Evaluation of Extracellular Matrix Protein Production Associated with Hyaline Cartilage in Organoids

To determine whether the generated organoids replicate the structure and composition of hyaline cartilage, the production of extracellular matrix (ECM) proteins was analyzed via immunohistochemistry (IHC) under different experimental conditions. The expression of aggrecan, type II collagen, and proteoglycan-4 (PRG4)—hallmark hyaline cartilage markers—was assessed, as well as type I and type X collagen, which are associated with hypertrophic cartilage. For expression controls, the following tissues were used: Negative Control (for all proteins): Adipose tissue; Positive Controls: Fetal trachea (for aggrecan, type II collagen, and PRG4), skin (for type I collagen), and intervertebral disc (for type X collagen) (Appendix A). The chondrocyte control line exhibited aggrecan, type II collagen, and PRG4 production, along with low levels of type I and type X collagen, consistent with its chondrogenic lineage and absence of hypertrophy markers (Appendix A and Figure 13G,H). SSC-derived organoids (+IM) displayed a more homogeneous structure compared to non-induced organoids (−IM) (Figure 13A). Under +IM conditions, there was a significant increase in aggrecan production, along with a moderate increase in type II and type X collagen. In contrast, −IM organoids did not produce type X collagen. However, PRG4 production showed no significant differences between conditions (Figure 13A,D). MSC-derived organoids exhibited very low levels of aggrecan and type II collagen but showed higher PRG4 and type X collagen production when comparing +IM and −IM conditions (Figure 13B,E). SSC-MSC co-culture organoids displayed high levels of aggrecan and PRG4, along with low levels of type II and type X collagen (Figure 13C,F).

## 4. Discussion

The results of this study provide valuable insights into the characterization and functional behavior of skeletal stem cells (SSCs) and mesenchymal stem cells (MSCs) derived from human bone marrow, as well as their capacity to form hyaline cartilage organoids under cartilage physioxia conditions. This discussion focuses on three main aspects: (i) initial characterization of the cell populations; (ii) formation and characterization of organoids, including molecular and functional assessments related to hyaline cartilage; and (iii) the potential role of PRG4 and PDPN as key markers for evaluating the chondrogenic quality of organoids.

The phenotypic and immunophenotypic characterization of SSCs was consistent with previous reports, showing expression of CD73, CD105, CD164, and PDPN, with the absence of CD31, CD34, CD45, and CD146. These cells exhibited high colony-forming capacity (up to 22 CFUs) and demonstrated strong chondrogenic and osteogenic potential, but limited adipogenic differentiation, reflecting their preference for the osteochondral lineage. In contrast, MSCs displayed a classical mesenchymal phenotype, expressing CD146, a marker associated with greater plasticity and hypertrophic potential. Their colony-forming capacity was lower (17 CFUs), and they exhibited multipotent differentiation potential, including chondrogenesis, osteogenesis, and adipogenesis [11,30,31,33,34].

The SSC-MSC co-culture allowed for the exploration of potential synergies between both populations, revealing a positive effect on chondrogenesis, with higher glycosaminoglycan production and increased *SOX9* expression, surpassing the levels observed in individual cell pools. This interaction suggests possible paracrine signaling and partial recreation of a physiological niche. Additionally, a reduction in adipogenesis was observed, indicating preferential commitment toward chondrogenesis, with relevant implications for cartilage engineering applications. Organoids generated under cartilage physioxia conditions exhibited high viability (>90%), maintained diameters close to 600 μm, and underwent progressive compaction. This greater cohesion was more evident in SSC-derived and SSC-MSC co-culture organoids, likely due to increased extracellular matrix (ECM) protein production and differences in cell adhesion molecule expression, although this remains to be confirmed. Molecular evaluation of these organoids revealed a gene expression profile similar to that of chondrocytes, with *SOX9*, *ACAN*, *PDPN*, *RUNX2*, and *COL10* expression, along with notable PRG4 expression. Protein detection further demonstrated that, particularly in SSC-derived and SSC-MSC co-culture organoids, high levels of aggrecan and PRG4 were produced, with low type X collagen expression. In contrast, MSC-derived organoids exhibited a more hypertrophic protein profile, with higher *RUNX2* and type X collagen expression, suggesting a greater tendency toward calcification.

The results of this study suggest that PDPN expression may play a key role in the structural stability and maintenance of an immature chondrogenic phenotype in SSC-derived organoids and SSC-MSC co-culture organoids. Although chondrogenic induction did not significantly increase PDPN expression at the gene level, it did enhance PDPN protein expression, which may be related to better organoid compaction, higher aggrecan production, and elevated *SOX9* expression. In contrast, MSC-derived organoids, which exhibited lower PDPN expression, showed reduced three-dimensional stability, absence of aggrecan, higher type X collagen production, and greater *RUNX2* expression, reflecting a tendency toward hypertrophy. These findings highlight PDPN as a functional marker associated with the chondrogenic quality of organoids, with the potential to promote a phenotype compatible with immature hyaline cartilage, particularly in SSC and SSC-MSC co-culture organoids under conditions simulating the physiological articular microenvironment. Similarly, PRG4 emerged as a relevant marker for assessing chondrogenic organoid quality, given its role in cartilage lubrication and homeostasis, as well as its hypertrophy-inhibitory function [53,54,55]. High PRG4 expression in SSC-derived and SSC-MSC co-culture organoids, combined with its low expression in MSC-derived organoids, reinforces its potential as a functional predictor for organoids with regenerative potential.

Future studies should evaluate these organoids via in vivo models under relevant biomechanical conditions, allowing for a deeper investigation of the functional roles of PRG4 and PDPN during cartilage defect regeneration. Additionally, it will be essential to analyze the transcriptomic and proteomic profiles before and after implantation, as well as assess tribological properties (friction and lubrication), to further establish these organoids as a viable tissue engineering strategy. Finally, integrating in vitro mechanical stress simulations could provide key insights into cellular adaptation to the native articular environment, allowing for further optimization of the functional chondrogenic organoid generation process.

## 5. Conclusions

This study demonstrates that skeletal stem cells (SSCs) and mesenchymal stem cells (MSCs) derived from human bone marrow exhibit distinct phenotypic, functional, and chondrogenic profiles, with SSCs showing a preferential commitment toward the osteochondral lineage and MSCs displaying broader multipotency. Under cartilage physioxia conditions, SSC-derived and SSC–MSC co-culture organoids exhibited superior chondrogenic features, including higher aggrecan and PRG4 production, greater structural cohesion, and reduced hypertrophy compared to MSC-derived organoids. The consistent association of PDPN expression with enhanced organoid stability and an immature hyaline cartilage phenotype highlights its potential as a structural and functional quality marker. Similarly, elevated PRG4 levels in SSC-containing organoids reinforce its relevance as a predictor of regenerative potential, owing to its roles in cartilage lubrication, homeostasis, and hypertrophy inhibition. Together, these findings support the use of SSCs, alone or in combination with MSCs, as promising cellular sources for generating high-quality chondrogenic organoids. Future in vivo studies under relevant biomechanical conditions, combined with transcriptomic, proteomic, and tribological analyses, will be critical for validating their therapeutic applicability in cartilage defect regeneration.

## Figures and Tables

**Figure 1 cells-14-01423-f001:**
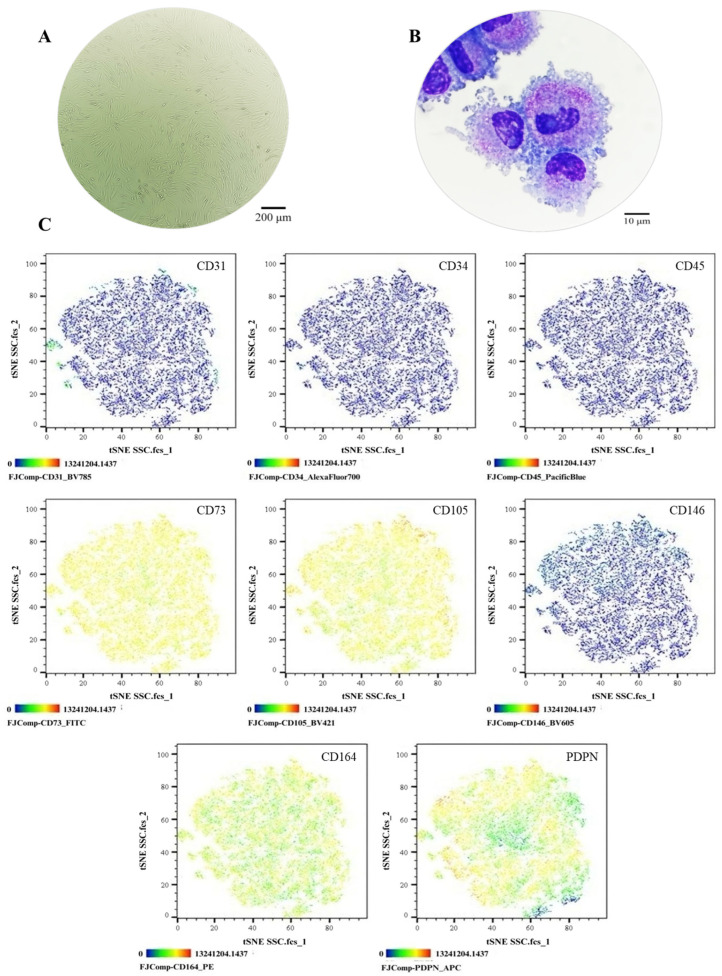
Phenotypic characterization of the SSC pool: (**A**) Cell morphology in culture, Olympus inverted microscope (10×). (**B**) Cell morphology in cytospin, ZEISS (Oberkochen, Germany) Axiolab 5 microscope (100×). (**C**) Immunophenotype analysis by spectral flow cytometry is displayed as a heatmap using t-SNE dimensionality reduction. The antigen assessed is indicated in the top right corner of each panel, and the color scale in the bottom left corner denotes antigen expression levels: negative (blue), weakly positive (green), moderately positive (yellow), and strongly positive (red).

**Figure 2 cells-14-01423-f002:**
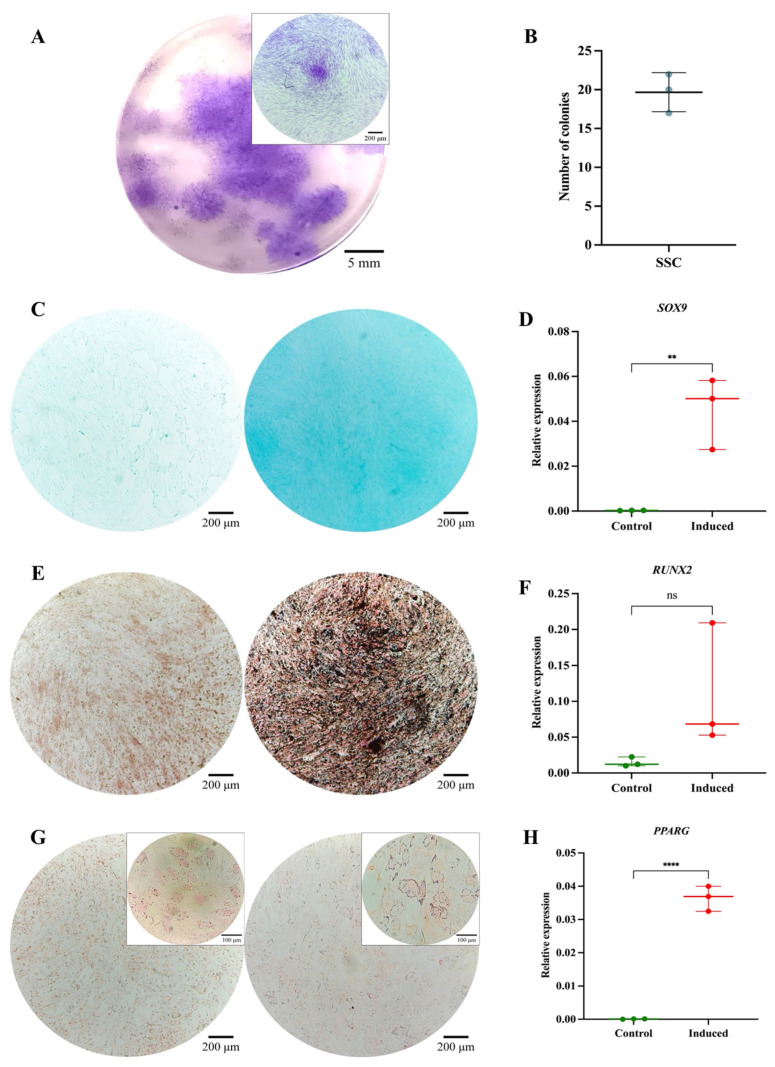
Functional characterization of the SSC pool: (**A**) Colony-forming unit (CFU) assay. Representative images of SSC-derived colonies after 14 days of culture, stained with 3% crystal violet. Images were acquired using an Olympus inverted microscope at 10× magnification (inset, 40×). (**B**) Absolute quantification of SSC-derived colonies after 14 days of culture. (**C**) Chondrogenic differentiation. SSCs cultured without (**left**) or with (**right**) chondrogenic differentiation medium. Alcian Blue staining at 10× magnification. (**D**) *SOX9* gene expression in undifferentiated (control) and differentiated (induced) SSCs (*p* = 0.0082). (**E**) Osteogenic differentiation. SSCs cultured without (**left**) or with (**right**) osteogenic differentiation medium. Von Kossa staining at 10× magnification. (**F**) *RUNX2* gene expression in undifferentiated (control) and differentiated (induced) SSCs (*p* = 0.1287). (**G**) Adipogenic differentiation. SSCs cultured without (**left**) or with (**right**) adipogenic differentiation medium. Oil Red O staining at 10× magnification (inset, 40×). (**H**) *PPARγ* gene expression in undifferentiated (control) and differentiated (induced) SSCs (*p* = 0.0082). Statistical significance was assessed using a two-tailed Student’s *t*-test. ** *p* < 0.01; **** *p* < 0.0001. ns: non-significant.

**Figure 3 cells-14-01423-f003:**
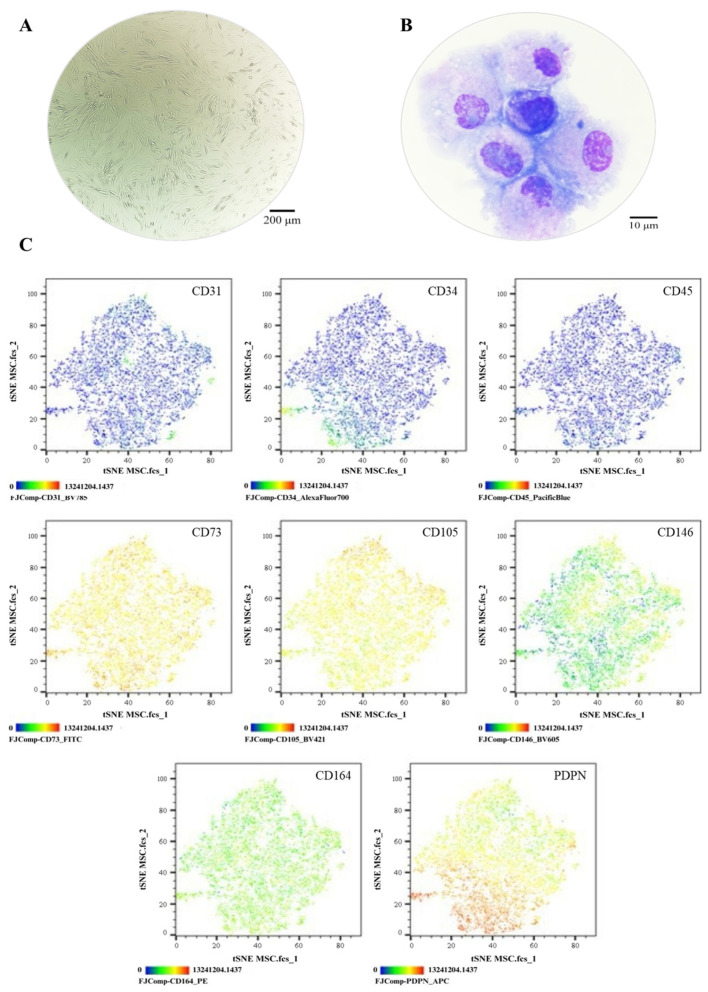
Phenotypic characterization of MSCs: (**A**) Cell morphology in culture, Olympus inverted microscope (10×). (**B**) Cell morphology in cytospin, ZEISS Axiolab 5 microscope (100×). (**C**) Immunophenotype analysis by spectral flow cytometry is displayed as a heatmap using t-SNE dimensionality reduction. The antigen assessed is indicated in the top right corner of each panel, and the color scale in the bottom left corner denotes antigen expression levels: negative (blue), weakly positive (green), moderately positive (yellow), and strongly positive (red).

**Figure 4 cells-14-01423-f004:**
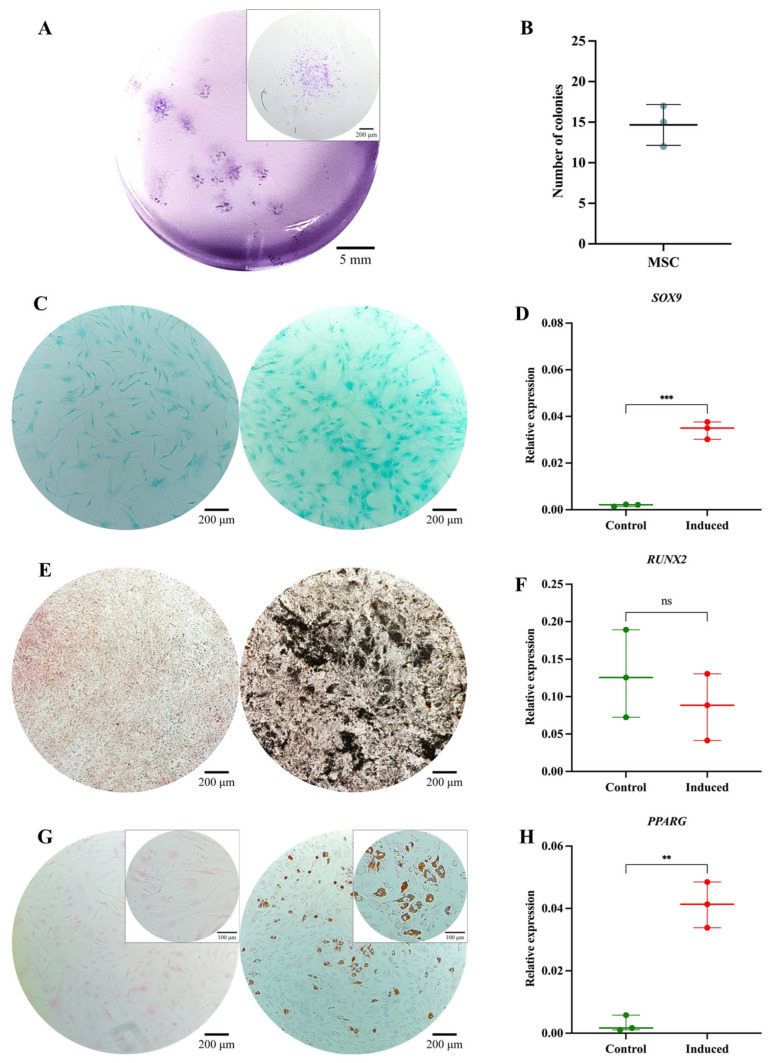
Functional characterization of the MSC pool: (**A**) Colony-forming unit (CFU) assay. Representative images of MSC-derived colonies after 14 days of culture, stained with 3% crystal violet. Images were acquired using an Olympus inverted microscope at 10× magnification (inset, 40×). (**B**) Absolute quantification of MSC-derived colonies after 14 days of culture. (**C**) Chondrogenic differentiation. MSCs cultured without (left) or with (right) chondrogenic differentiation medium. Alcian Blue staining at 10× magnification. (**D**) *SOX9* gene expression in undifferentiated (control) and differentiated (induced) MSCs (*p* = 0.0001). (**E**) Osteogenic differentiation. MSCs cultured without (**left**) or with (**right**) osteogenic differentiation medium. Von Kossa staining at 10× magnification. (**F**) *RUNX2* gene expression in undifferentiated (control) and differentiated (induced) MSCs (*p* = 0.3744). (**G**) Adipogenic differentiation. MSCs cultured without (**left**) or with (**right**) adipogenic differentiation medium. Oil Red O staining at 10× magnification (inset, 40×). (**H**) *PPARγ* gene expression in undifferentiated (control) and differentiated (induced) MSCs (*p* = 0.0010). Statistical significance was assessed using a two-tailed Student’s *t*-test. ** *p* < 0.01; *** *p* < 0.001. ns: non-significant.

**Figure 5 cells-14-01423-f005:**
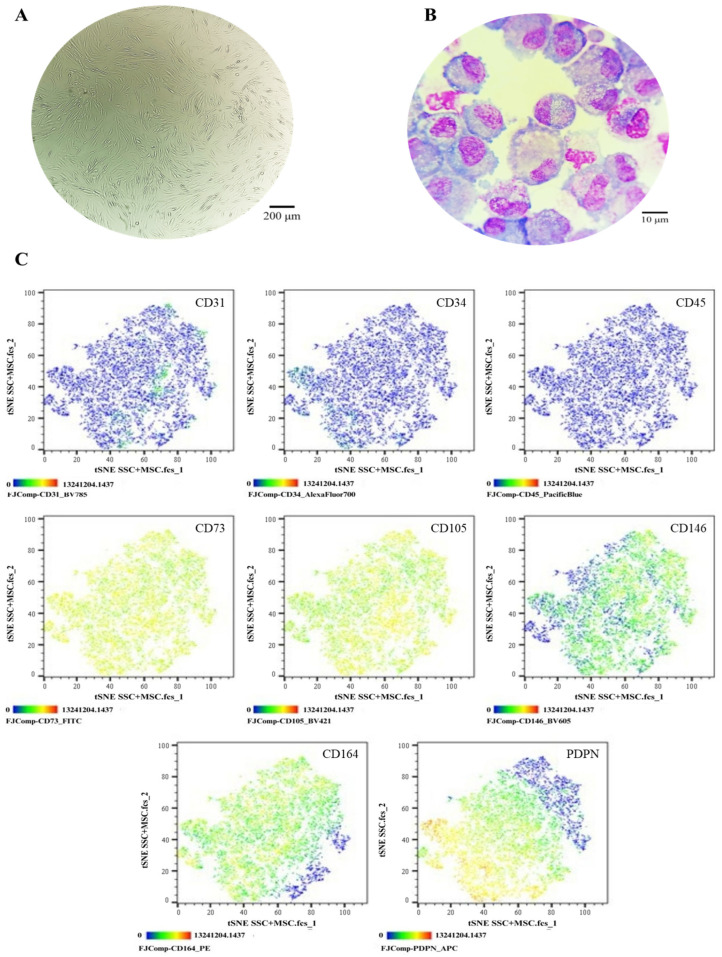
Phenotypic characterization of SSCs in co-culture with MSCs: (**A**) Cell morphology in culture, Olympus inverted microscope (10×). (**B**) Cell morphology in cytospin, ZEISS Axiolab 5 microscope (100×). (**C**) Immunophenotype analysis by spectral flow cytometry is displayed as a heatmap using t-SNE dimensionality reduction. The antigen assessed is indicated in the top right corner of each panel, and the color scale in the bottom left corner denotes antigen expression levels: negative (blue), weakly positive (green), moderately positive (yellow), and strongly positive (red).

**Figure 6 cells-14-01423-f006:**
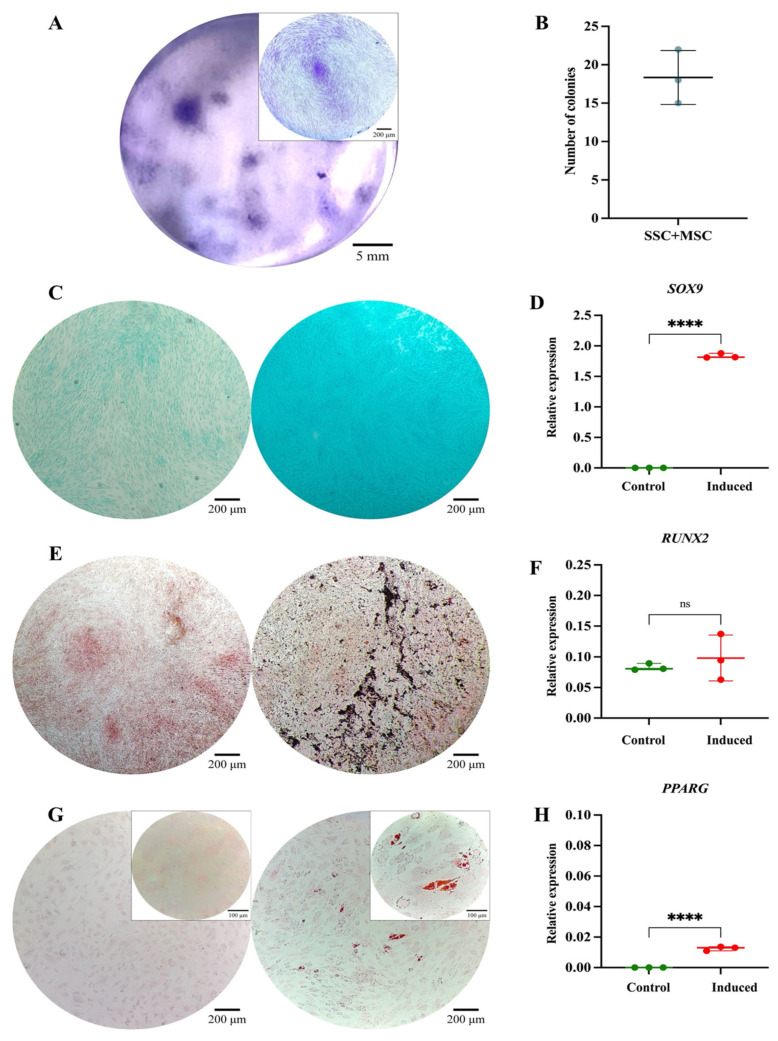
Functional characterization of the SSC pool in co-culture with MSCs: (**A**) Colony-forming unit (CFU) assay. Representative images of SSC-MSC co-culture-derived colonies after 14 days of culture, stained with 3% crystal violet. Images were acquired using an Olympus inverted microscope at 10× magnification (inset, 40×). (**B**) Absolute quantification of SSC-MSC co-culture-derived colonies after 14 days of culture. (**C**) Chondrogenic differentiation. SSC-MSC co-culture without (**left**) or with (**right**) chondrogenic differentiation medium. Alcian Blue staining at 10× magnification. (**D**) *SOX9* gene expression in undifferentiated (control) and differentiated (induced) MSCs (*p* = 0.0001). (**E**) Osteogenic differentiation. MSCs cultured without (**left**) or with (**right**) osteogenic differentiation medium. Von Kossa staining at 10× magnification. (**F**) *RUNX2* gene expression in undifferentiated (control) and differentiated (induced) MSCs (*p* = 0.5282). (**G**) Adipogenic differentiation. MSCs cultured without (**left**) or with (**right**) adipogenic differentiation medium. Oil Red O staining at 10× magnification (inset, 40×). (**H**) *PPARγ* gene expression in undifferentiated (control) and differentiated (induced) MSCs (*p* = 0.0001). Statistical significance was assessed using a two-tailed Student’s *t*-test. **** *p* < 0.0001. ns: non-significant.

**Figure 7 cells-14-01423-f007:**
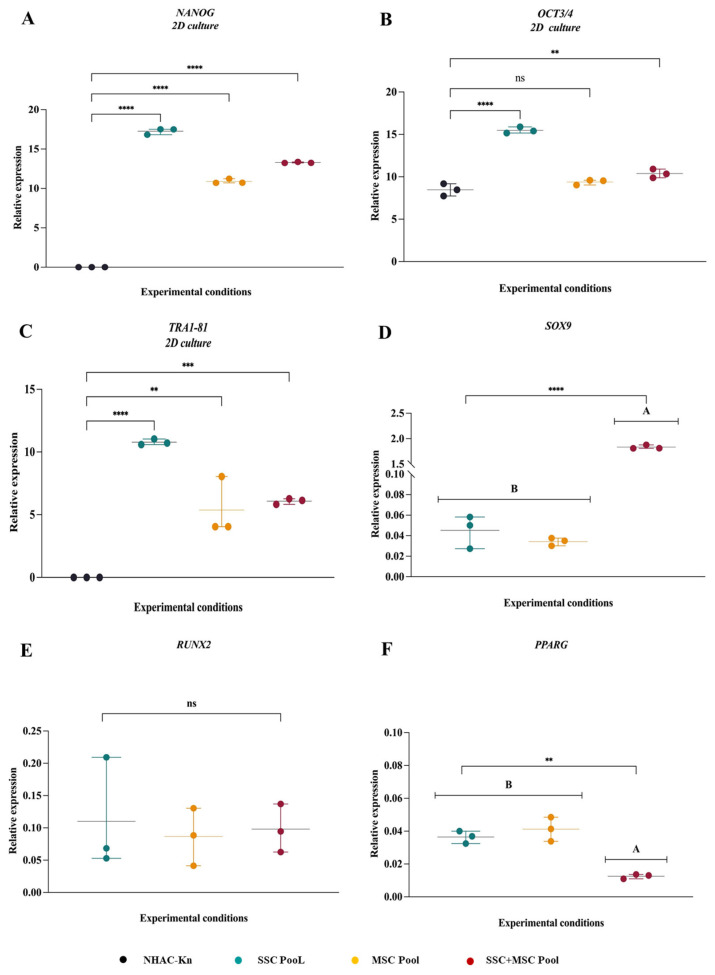
Comparative gene expression profiles associated with pluripotency and multipotency in SSC, MSC, and SSC–MSC co-culture pools: (**A**–**C**) Relative expression levels of pluripotency-associated genes (*NANOG*, *OCT3/4*, and *TRA1-81*) in SSC, MSC, and SSC–MSC co-culture pools cultured in basal medium (i.e., without differentiation stimuli). (**D**–**F**) Expression of lineage-specific transcription factors related to multipotency—*SOX9* (chondrogenesis), *RUNX2* (osteogenesis), and *PPARγ* (adipogenesis)—in cell pools subjected to lineage-specific differentiation media. The statistical difference is generated by Group A. Statistical significance was assessed using one-way ANOVA followed by post hoc multiple comparisons; ** *p* < 0.01; *** *p* < 0.001; **** *p* < 0.0001, ns: non-significant.

**Figure 8 cells-14-01423-f008:**
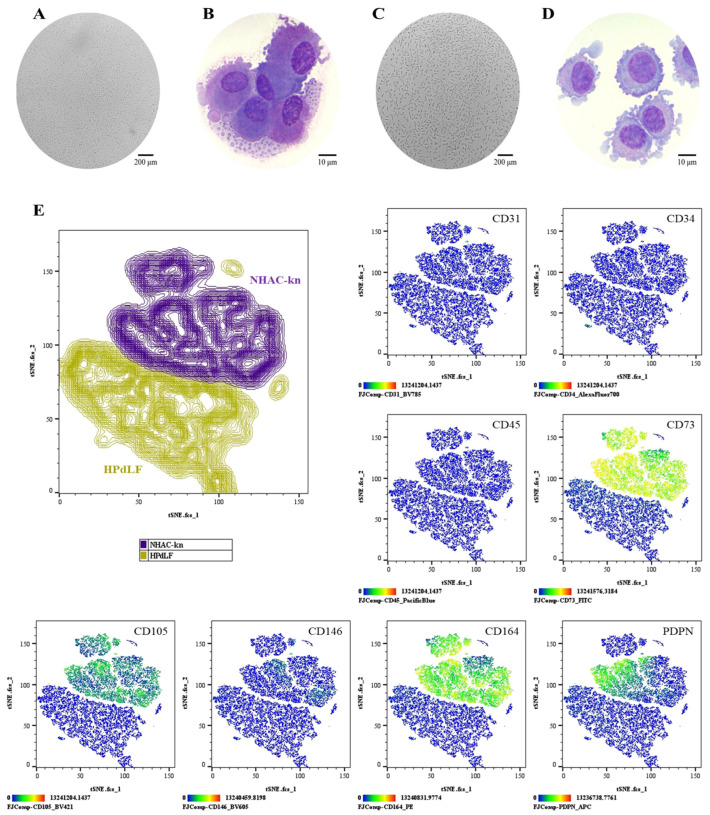
Comparative phenotypic characterization of control cell populations: (**A**,**B**) NHAC-kn: Cell morphology in culture (Olympus inverted microscope, 10×) and in cytospin preparations (ZEISS Axiolab 5 microscope, 100×). (**C**,**D**) HPdLF: Cell morphology in culture (Olympus inverted microscope, 10×) and in cytospin preparations (ZEISS Axiolab 5 microscope, 100×). (**E**) t-SNE analysis was performed to visualize the spatial segregation of the two control cell populations, providing a reference framework for the interpretation of subsequent immunophenotyping plots. Immunophenotypic profiling by spectral flow cytometry, displayed as heatmaps generated using t-SNE dimensionality reduction. The antigen analyzed is indicated in the top right corner of each panel. The color scale in the bottom left indicates expression levels: negative (blue), weakly positive (green), moderately positive (yellow), and strongly positive (red).

**Figure 9 cells-14-01423-f009:**
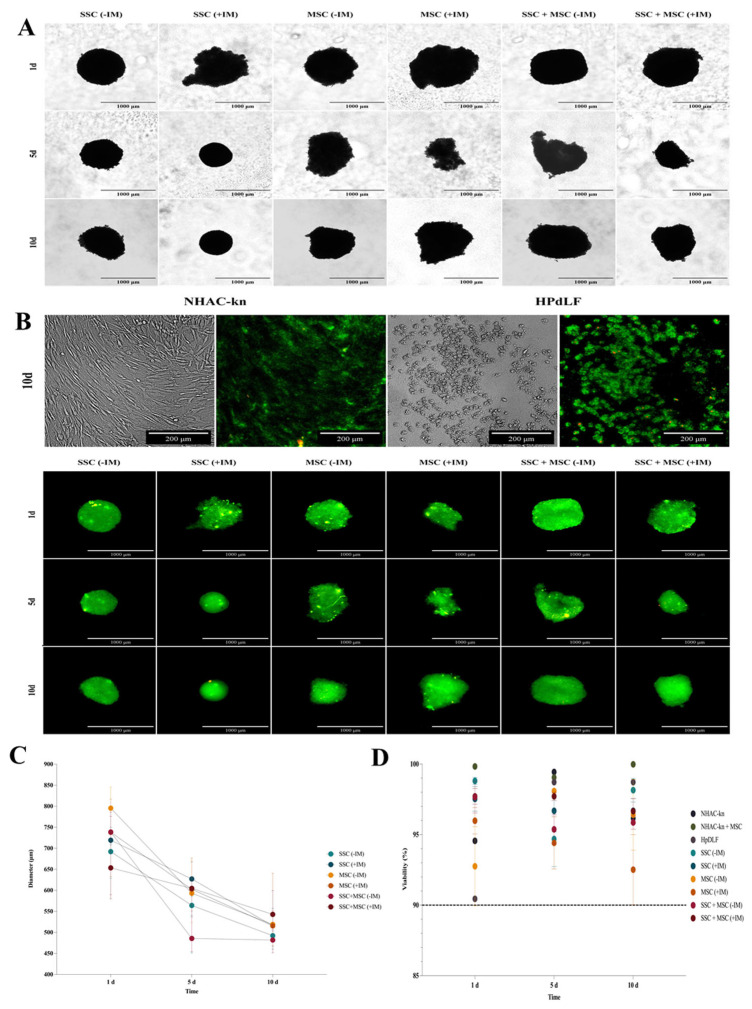
Diameter and viability assessment of cartilage organoids derived from co-cultured SSC, MSC, and SSC–MSC groups: (**A**) Representative images of cartilage organoids at days 1, 5, and 10 of culture, derived from SSC, MSC, and SSC–MSC groups, cultured with or without chondrogenic differentiation medium. Images were acquired using a Cytation 5 imaging system (BioTek) under 10× magnification. (**B**) Viability assessment of spheroids derived from control cell populations (NHAC-kn and HPdLF) and from SSC, MSC, and SSC–MSC groups, cultured with or without chondrogenic differentiation medium at days 1, 5, and 10. Cell viability was evaluated using the LIVE/DEAD™ Cell Imaging Kit (Invitrogen, Thermo Fisher Scientific^®^). Live cells fluoresce green (calcein AM), and dead cells fluoresce red (ethidium homodimer-1). Images were acquired with the Cytation 5 system at 10× magnification. (**C**) Quantification of spheroid diameters at days 1, 5, and 10 of culture. (**D**) Percentage viability of control cell populations and cartilage organoids at days 1, 5, and 10. The dashed line represents the 90% viability threshold, indicating that all populations maintained >90% viability under the experimental conditions throughout the culture period. Data on diameters and viability under the experimental conditions were acquired and analyzed using the Cytation 5 imaging system (BioTek).

**Figure 10 cells-14-01423-f010:**
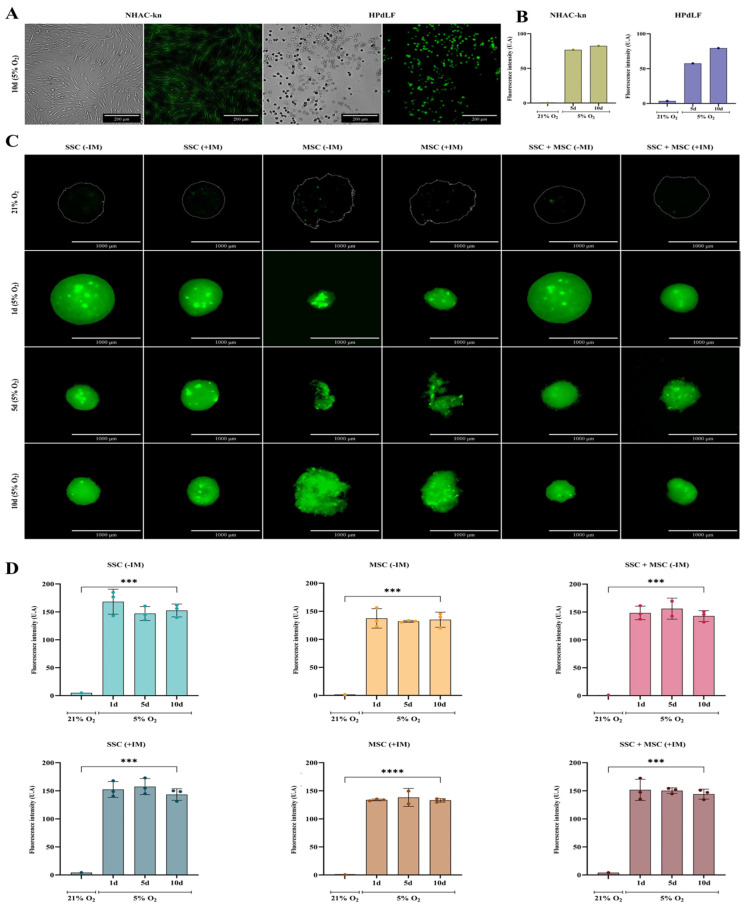
Assessment of oxygen levels in cartilage organoids and control cell populations: (**A**) Representative images of control cell lines (NHAC-kn and HPdLF) after 10 days of culture under 5% O_2_. Bright-field and fluorescence images were acquired following staining with the hypoxia-sensitive probe BioTracker™ 520 Green Hypoxia Dye (Sigma-Aldrich^®^). Under normoxic conditions (21% O_2_), the dye remains inactive or exhibits low fluorescence. Under hypoxic conditions (5% O_2_), it is chemically reduced by intracellular enzymes, such as oxidoreductases, resulting in green fluorescence activation. (**B**) Quantification of fluorescence intensity in control cell populations cultured under 21% and 5% O_2_. Increased fluorescence intensity indicates lower oxygen availability. (**C**) Fluorescence and bright-field images of cartilage organoids derived from SSC, MSC, and SSC–MSC co-culture pools, cultured with or without chondrogenic differentiation medium. Organoids were cultured at 21% O_2_ (no fluorescence) or 5% O_2_ for 1, 5, and 10 days, showing increasing green fluorescence under hypoxic conditions. (**D**) Quantification of fluorescence intensity in cartilage organoids cultured under 21% and 5% O_2_. Significantly higher fluorescence was observed at 5% O_2_. Statistical significance was determined by one-way ANOVA with post hoc multiple comparisons; *** *p* < 0.001; **** *p* < 0.0001.

**Figure 11 cells-14-01423-f011:**
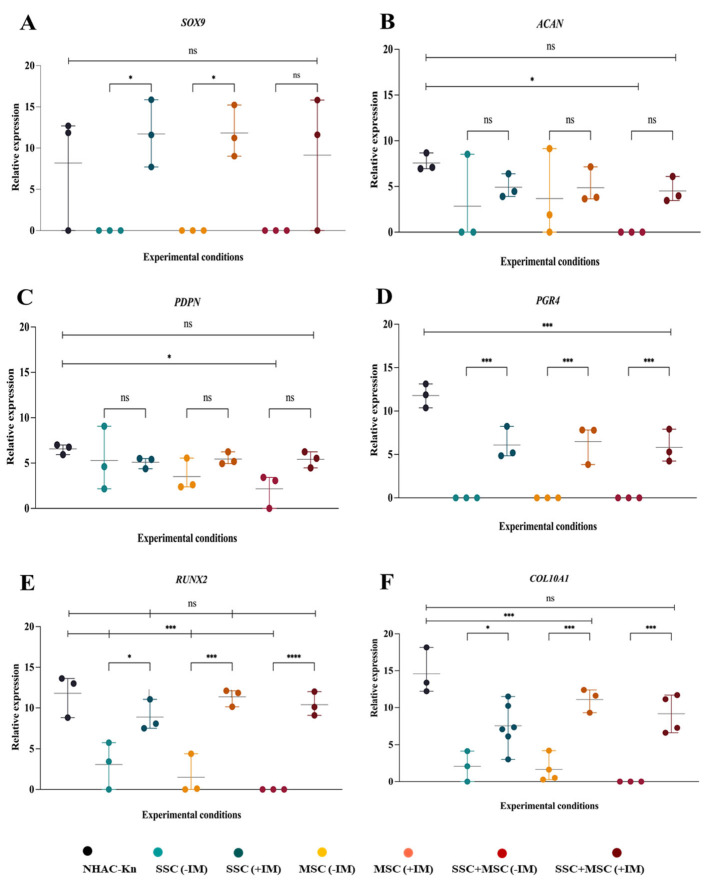
Expression of genes associated with chondrogenesis in cartilage organoids derived from SSC, MSC, and SSC–MSC co-culture pools after 10 days of culture: (**A**–**D**) Relative expression of *SOX9*, *ACAN*, *PDPN*, and PRG4 in chondrocytes from the NHAC-kn cell line and in cartilage organoids cultured with or without chondrogenic differentiation medium. (**E**,**F**) Relative expression of *RUNX2* and *COL10A1* in chondrocytes from the NHAC-kn cell line and in cartilage organoids under the same conditions. Statistical significance was assessed using a two-tailed Student’s *t*-test; * *p* < 0.05; *** *p* < 0.001; **** *p* < 0.0001. ns: non-significant.

**Figure 12 cells-14-01423-f012:**
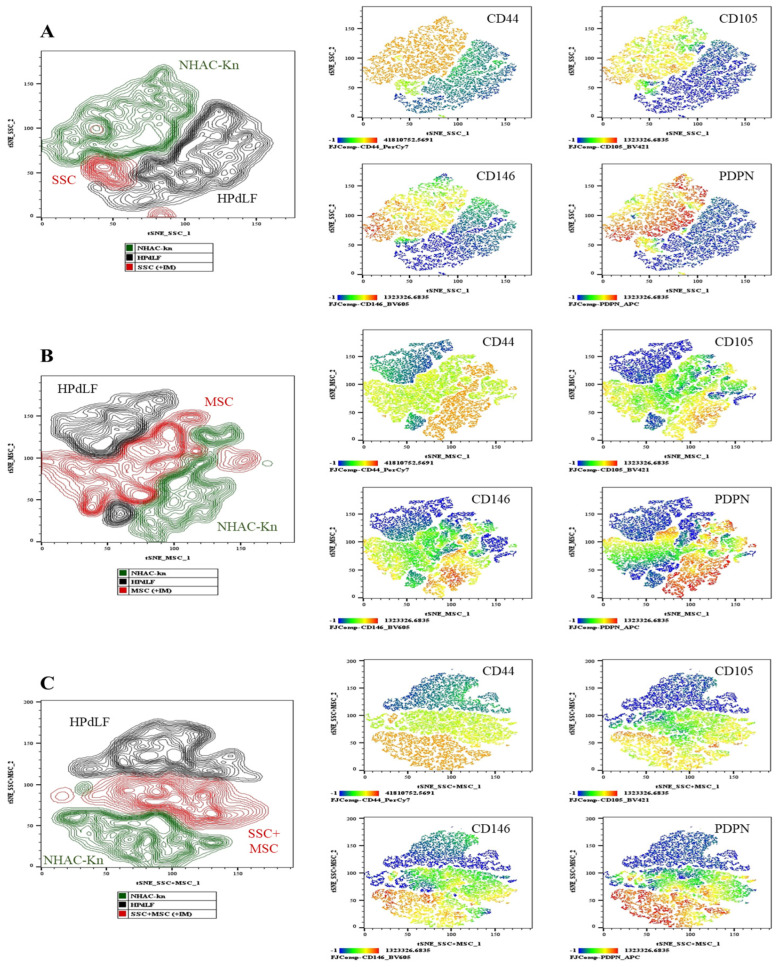
Generation of chondrocytes from cartilage organoids derived from SSCs, MSCs, and SSC–MSC co-cultures: After 10 days of culture in chondrogenic differentiation medium, cartilage organoids were enzymatically disaggregated, and the generation of chondrocytes exhibiting a CD44^+^/CD105^+^/CD146^+^/PDPN^+^ phenotype was assessed by spectral flow cytometry. The immunophenotypic profiles of organoid-derived cells were compared with those of control cell lines (NHAC-kn and HPdLF). (**A**) The t-SNE analysis illustrating the spatial distribution of the two control populations and SSC-derived organoid cells. Heatmaps and t-SNE plots show the expression levels of CD44, CD105, CD146, and PDPN. (**B**) The t-SNE analysis illustrating the spatial distribution of the control populations and MSC-derived organoid cells. Heatmaps and t-SNE plots show the expression of the same antigenic markers. (**C**) The t-SNE analysis illustrating the spatial distribution of the control populations and cells derived from SSC–MSC co-culture organoids. Heatmaps and t-SNE plots indicate expression of CD44, CD105, CD146, and PDPN. Statistical significance was assessed using one-way ANOVA followed by multiple comparisons. The antigen analyzed is indicated in the top right corner of each panel. The color scale in the bottom left indicates expression levels: negative (blue), weakly positive (green), moderately positive (yellow), and strongly positive (red).

**Figure 13 cells-14-01423-f013:**
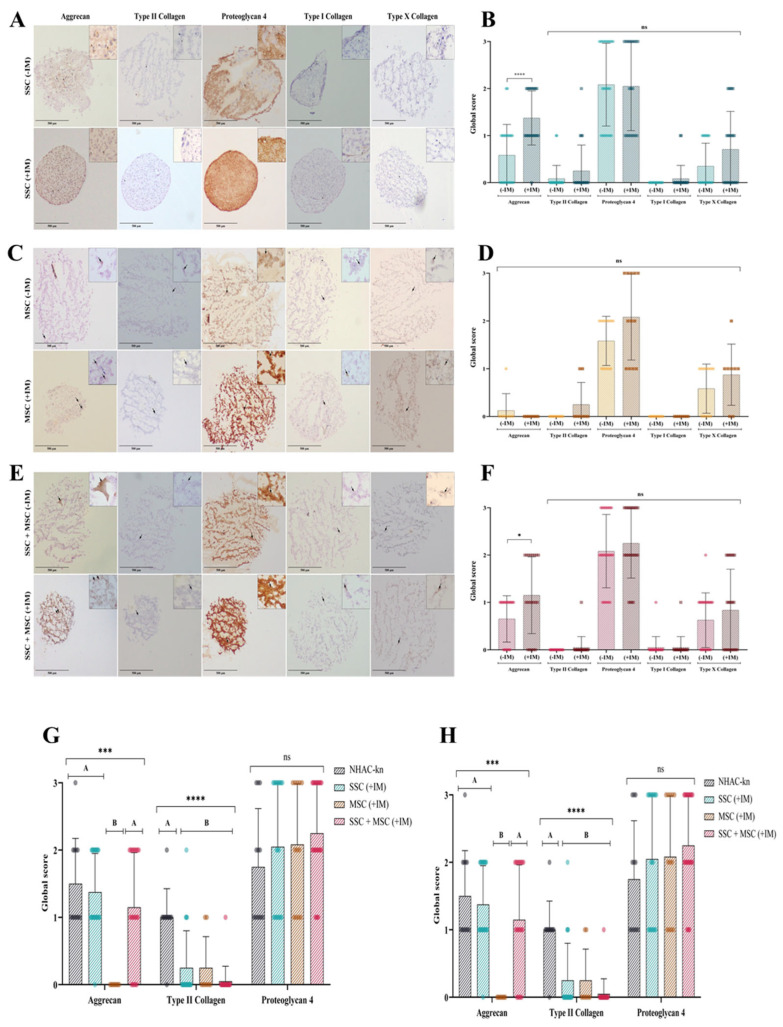
Extracellular matrix (ECM) protein production in cartilage organoids: (**A**) Immunohistochemical detection of aggrecan, type II collagen, proteoglycan-4, type I collagen, and type X collagen in SSC-derived organoids cultured without (–IM) or with (+IM) chondrogenic induction medium. Images acquired with an Olympus microscope at 10× magnification. (**B**) Immunohistochemical detection of the same ECM proteins in MSC-derived organoids under the same conditions. (**C**) Immunohistochemical detection of ECM proteins in organoids derived from SSC–MSC co-cultures, with or without chondrogenic induction medium. Images acquired with an Olympus microscope at 10× magnification. (**D**–**F**) Semi-quantitative scoring of ECM protein expression (aggrecan, type II collagen, proteoglycan-4, type I collagen, and type X collagen) in SSC-derived organoids (**D**), MSC-derived organoids (**E**), and SSC–MSC co-culture organoids (**F**), cultured with or without chondrogenic induction medium. (**G**) Comparative scoring of aggrecan, type II collagen, and proteoglycan-4 expression between cartilage organoids and the NHAC-kn chondrocyte line. (**H**) Comparative scoring of type I and type X collagen expression between cartilage organoids and the NHAC-kn chondrocyte line. The statistical difference is generated by Group A (**G**,**H**). Statistical significance was determined using the Kruskal–Wallis test followed by multiple comparisons; * *p* < 0.05; *** *p* < 0.001; **** *p* < 0.0001. ns: non-significant.

**Table 1 cells-14-01423-t001:** Donor characteristics and bone marrow samples for SSC and MSC isolation.

	Sample No.	Age (Years)	Gender (M: Male/F: Female)	BM Volume (mL)
SSC	01	55	M	90
	02	61	M	65
	03	69	M	42
	04	59	M	95
	05	73	F	75
	06	68	F	55
MSC	07	79	M	90
	08	61	M	75
	09	83	F	80
	10	73	M	90
	11	84	F	80
	12	60	M	80

## Data Availability

The data related to the voluntary bone marrow donors who attended the Department of Orthopedics and Traumatology at the Hospital Universitario San Ignacio are confidential and cannot be publicly disclosed to protect donor privacy. However, the anonymized data corresponding to the analyzed variables are available from the corresponding author upon reasonable request. Informed consent was obtained from all subjects involved in the study.

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
