# Peer review of "Interaction Between Human Skeletal and Mesenchymal Stem Cells Under Physioxia Enhances Cartilage Organoid Formation: A Phenotypic, Molecular, and Functional Characterization"

_cells, 2025, doi:10.3390/cells14181423_

Round 1
Reviewer 1 Report
Comments and Suggestions for Authors This manuscript investigates the interaction between human skeletal stem cells (SSCs) and mesenchymal stem cells (MSCs) under physiologic oxygen tension (physioxia) in the context of cartilage organoid formation. The study is well-structured, employs rigorous methodology, and addresses an important gap in cartilage tissue engineering. The study is interesting, however, there are several points need to be drawn before the paper can be accepted for publication: 1. The osteogenic, chondrogenic, and adipogenic differentiation results are currently presented qualitatively. The authors should include quantitative measurements. For reference, osteogenesis can be quantified at OD 405 nm, and adipogenesis at OD 450 nm. 2. Skeletal stem cells have shown promising potential for treating bone diseases, with several important papers published recently, including Ambrosi et al. (Cell Stem Cell, 2025). The authors should include this in the discussion.Author Response
Dear Reviewer,
We sincerely appreciate your valuable and constructive evaluation of our manuscript. Please find attached a PDF file containing our detailed responses to each of your comments.
Thank you for your time and insightful feedback, which has helped us improve the quality of our work.

Reviewer 2 Report
Comments and Suggestions for Authors
The paper entitled “Interaction Between Human Skeletal and Mesenchymal Stem Cells Under Physioxia Enhances Cartilage Organoid Formation: A Phenotypic, Molecular, and Functional Characterization” by Azain deals with the topic of cartilage regeneration. It attempts to show that human skeletal stem cells can also undergo chondrogenic differentiation and can be used as a potential source of regeneration. The study is very comprehensive and attempts to demonstrate the differentiation capacity and characteristics under different stimulations and conditions. Based on their results, the authors conclude that SSCs represent a potential source. The introduction adequately introduces the topic and provides the necessary information to understand the study. The reviewer greatly appreciates that the authors have formulated a clear hypothesis. The materials and methods section is described in a clear and comprehensible manner.
The results build on each other and the text is correctly formulated in relation to the graphics presented. The discussion could be more closely linked to the current literature and should not merely reflect the authors' own results.
Overall, it is an interesting study on a topic that is still highly relevant. However, the authors should revise a few points before publication:
- How do the authors justify the statement in the title: ...functional characterization...? What are the authors basing this on?
- No abbreviations should remain unexplained in the abstract. An abstract should be designed in such a way that it is easily understandable to the general public. Therefore, the reviewer would recommend that the authors explain all abbreviations the first time they appear or, if the word count is insufficient, focus on the most striking results.
- The reviewer kindly asks the authors to check the entire document for the following error: induction medium (+IM) according to the reviewer's understanding, and then suddenly the abbreviations (MI) appear again and again! This must be consistent throughout the document!!!
- Line 52: The reviewer would suggest not using the term “consists of a single cell population” as there may also be chondrogenic progenitor stem cells.
- Line 68: Why do the authors write MSC and SSCs and not, as in the title, SSCs first and then MSCs? Here, too, the reviewer would greatly appreciate greater consistency.
- Lines 152, 158, 164: The reviewer asked the authors to carefully revise the presentation of size specifications.
- Line 18: Typo “MSCs, and an MSC-SSC...” Correct to “MSCs, and a MSC-SSC co-culture.”
- Line 247: O2: Corrected to subscript 2
- - A total of 6 SSCs and MSCs donors are listed in Table 1, but why were only N=3 used in the experiments? This is a pity, as it could have improved the significance of the entire study.
- In Fig. 7: Why are the NHAC data missing from graphs D, E, and F?
- Fig. 9 C: Why does the x-axis suddenly show 1, 2, 3 days? Shouldn't it correctly show 1, 5, 10 days, as described in the image description?
- Fig. 11: Here, too, the legend with the colored circles is incorrectly labeled: -MI and +MI should correctly be labeled -IM and +IM.
All the best
Kind regards,
Author Response
Dear Reviewer,
We sincerely appreciate your valuable and constructive evaluation of our manuscript. Please find attached a PDF file containing our detailed responses to each of your comments.
Thank you for your time and insightful feedback, which has helped us improve the quality of our work.

Round 2
Reviewer 1 Report
Comments and Suggestions for Authors
accepted